# PS-AAS: Portfolio Selection for Automated Algorithm Selection in Black-Box Optimization

**Ana Kostovska**[1,2] **Gjorgjina Cenikj**[1,2] **Diederick Vermetten**[3] **Anja Jankovic**[4]
**Ana Nikolikj**[1,2] **Urban Škvorc**[1] **Peter Korošec**[1] **Carola Doerr**[4] **Tome Eftimov**[1]

[1]Jožef Stefan Institute, Ljubljana, Slovenia
[2]Jožef Stefan Postgraduate School, Ljubljana, Slovenia
[3]Leiden University, LIACS, Leiden, The Netherlands
[4]Sorbonne Université, LIP6, CNRS, Paris, France

**Abstract**  The performance of automated algorithm selection (AAS) strongly depends on the portfolio of algorithms to choose from. Selecting the portfolio is a non-trivial task that requires balancing the trade-off between the higher flexibility of large portfolios with the increased complexity of the AAS task. In practice, probably the most common way to choose the algorithms for the portfolio is a greedy selection of the algorithms that perform well in some reference tasks of interest.

We set out in this work to investigate alternative, data-driven portfolio selection techniques. Our proposed method creates algorithm behavior meta-representations, constructs a graph from a set of algorithms based on their meta-representation similarity, and applies a graph algorithm to select a final portfolio of diverse, representative, and non-redundant algorithms. We evaluate two distinct meta-representation techniques (SHAP and performance2vec) for selecting complementary portfolios from a total of 324 different variants of CMA-ES for the task of optimizing the BBOB single-objective problems in dimensionalities 5 and 30 with different cut-off budgets. We test two types of portfolios: one related to overall algorithm behavior and the 'personalized' one (related to algorithm behavior per each problem separately). We observe that the approach built on the performance2vec-based representations favors small portfolios with negligible error in the AAS task, whereas the portfolios built from the SHAP-based representations gain from higher flexibility at the cost of decreased performance of the AAS. Across most considered scenarios, personalized portfolios yield comparable or slightly better performance than the classical greedy approach. They outperform the full portfolio in all scenarios.

## 1 Introduction

Automated algorithm selection (AAS) has gained substantial traction in black-box optimization in recent years [KHNT19, KJV+22, MRW+22, MSKH15]. The choice of the best suited algorithm for a given black-box optimization problem is an important challenge, dependent on the properties of the problem instance at hand, as well as on available computational resources. In particular, the AAS problem becomes considerably more complex as the number of possible algorithms to choose from grows larger. In practice, we might have to deal with hundreds of different algorithms at a time. A typical AAS pipeline consists of (1) learning the mapping between (a) representations of *black-box problem instances* and (b) performances (or rankings) of different *optimization algorithms* on each of these instances (usually via a supervised machine learning (ML) model), then (2) using the model's output on a new problem instance to base the decision of which algorithm to use in a new scenario. Determining which problem instances and which algorithms to include for training of an AAS model are important questions, since the model's performance is crucially influenced by both aspects.

Selecting among black-box solvers for the algorithm portfolio in order to balance the trade-off between the flexibility of the portfolio (i.e., having more options to choose from) and the increasing complexity of the AAS task (incurred by the size of the portfolio) remains a largely neglected topic. This is particularly true when we task ourselves with choosing among a large family of similar algorithms, such as the modular CMA-ES variants [dNVW+21], where the full portfolio lacks diversity by design, rendering AAS all the more complex. We typically opt for one of the following two naïve baselines: either we consider as many algorithms as possible for which we have performance data on many different problems, or we pick those algorithms that are top-performing on some reference problems of interest (*greedy* portfolio selection, see [KT19] for an example). However, we observe a lack of works addressing alternative, data-driven techniques for portfolio selection, which might provide AAS performance gains, while offering further insights into the interplay between the composition of the portfolio and the AAS performance.

Our application of interest is AAS for black-box optimization. Different from classic AAS tasks such as for SAT or TSP solving, in black-box optimization, we do not have access to an explicit formulation of a problem instance. We can therefore only learn about problem instances by approximating them using a set of already evaluated solution candidates, for example via tools from exploratory landscape analysis (ELA) [MBT+11].

**Related work**: While numerous studies in black-box optimization are concerned with designing diverse and representative *problem instance* portfolios in an automated way [MS20, DM22, WCLW20], there is a surprising lack of works tackling automated *algorithm* portfolio selection in this domain. Algorithm portfolios have been studied almost exclusively as a term depicting "combining complementary strengths of multiple algorithms into an all-encompassing solver" [LHH15, BP14], but barely any works have so far analyzed data-driven, automated approaches to choose which algorithms to include in the final portfolio, particularly in the case where we have a huge initial number of algorithms at our disposal. One such rare work in classical optimization [WDS20] takes into consideration the concept of creating algorithm portfolios for resolving berth allocation problems (BAP) [BM15] with time budget restrictions, and shows that portfolios based on a data-driven selection of algorithms outperform those based on traditional algorithm performance statistics.

**Our contribution**: We investigate two data-driven approaches for portfolio selection for AAS in black-box optimization, based on two distinct meta-representations of algorithm behavior. We start with a large set of 324 modular CMA-ES variants [dNVW+21] used to solve the 24 noiseless single-objective BBOB problems [HFRA09]. We examine two different techniques for algorithm meta-representations, performance2vec- and SHAP-based, and we create two types of portfolios: one that takes into account overall (i.e., aggregated) algorithm behavior, and another that takes into account local algorithm behavior, i.e., personalized to each problem. We analyze the performance of our data-driven approaches on the AAS task for five different cut-off budgets (500, 2 000, 5 000, 10 000, 50 000) and two problem dimensions ($5D$ and $30D$). Our findings indicate that performance2vec-based portfolios are best suited for small portfolios with minimal error in the AAS task compared to the virtual best solver from the selected portfolio. On the other hand, portfolios built from SHAP-based meta-representations offer greater flexibility at the expense of AAS performance. In most cases, personalized portfolios perform similarly or slightly better than the greedy approach and outperform the full portfolio in all scenarios.

## 2  Data-driven Algorithm Portfolio Selection

Our data-driven algorithm portfolio selection consists of two steps: learning a meta-representation of algorithm performances and using it to select algorithms to include in the final portfolio.

**1. Learning meta-representations**: To learn the meta-representations, each algorithm is run on $k$ instances of $n$ problem classes $m$ independent times, due to the stochastic nature of the algorithms. The experimental outcomes are then represented in a $nk \times m$-dimensional matrix $Y$, wherein each row corresponds to a particular problem instance and each column corresponds to the quality of

the best solution found by the algorithm after a predetermined number of function evaluations. We use this data to compute two meta-representations of the algorithms:

***Performance2vec [EPKK20]:*** This meta-representation is defined using the information obtained in the performance space distributed across different problem classes. First, the quality of the solution of a given problem instance $j$ of the problem $i$ is computed as the median across multiple runs of the algorithm on that particular problem instance (i.e., $y_{(i-1)k+j} = \text{median}(Y_{(i-1)k+j,*})$, where $Y_{(i-1)k+j,*}$ is a row of the matrix $Y$. Next, for each algorithm, we compute the $n$-dimensional vector $p2v = (p2v_1, \ldots, p2v_n)$ with $p2v_i := \sum_{j=1}^{k} y_{(i-1)k+j}/k$, the average across median solution qualities per each instance from the problem $i$.

***Shapley:*** This meta-representation is an indirect outcome of performing explainable automated algorithm performance prediction. We typically explain model predictions using the SHAP method [LL17], which quantifies feature importance. This method assumes the input landscape features as players and the ML model performance as the payoff. Shapley values assigned to the landscape features measure the extent to which each feature contributes to the overall ML model ability to predict the quality of the solution of the algorithm when evaluated on a specific set of data points. Based on this, Shapley values can be treated as meta-representations of the algorithm behavior that capture interactions of the problem instance space and the algorithm performance space. SHAP is one of the most researched state-of-the-art feature selection methods that provides both global (i.e., on a set of problem instances) and local explanations (i.e., on a particular problem instance) [ME20].

In our setup, we assume that the landscape properties of the problem instances are represented with $p$ features, $f_q, q = 1, \ldots, p$. The feature vector representation of an instance $j$ ($j = 1, \ldots, k$) that belongs to the problem $i$ ($i = 1, \ldots, n$) is linked to the quality of the algorithm achieved for a fixed number of function evaluations (in our case, we use again median solution quality, $y_{(i-1)k+j}$). This allows us to organize the data into an $nk \times (p+1)$-dimensional matrix, where for each problem instance (represented as a row), the first $p$ columns are their landscape properties and the last column corresponds to algorithm performance. The data is split into train and test instance sets. We train an explainable single-target regression (ML) model, which takes as input the landscape features and predicts the quality of the solution achieved by the algorithm as output. The model is then evaluated on the test data instances. We compute Shapley values on the test set, and by averaging them across all problem instances for each landscape feature separately, we generate a novel $p$-dimensional algorithm meta-representation. These meta-representations have been previously used to identify algorithm behavior on a set of benchmark problem instances [NLKE22] and to predict from performance data the configuration of a modular CMA-ES framework [KVD+22b].

**2. Sampling heuristics.** The SELECTOR methodology [CLE+22] enables the selection of a subset of diverse, representative and non-redundant algorithms from a larger set described using meta-representations. To select the algorithms, we use the Maximal Independent Sets (MIS) and Dominating Sets (DS) algorithms. These approaches require the generation of a graph representation of the entire dataset, which is constructed such that the nodes represent the data instances, which are connected with an edge if the cosine similarity of their feature representations (i.e., meta-representations) exceeds a certain threshold. The MIS algorithm then selects a subset of the nodes in the graph that are not mutually connected by an edge, i.e., no two nodes in the selected subset have a meta-representation similarity that exceeds the predefined thresholds. On the other hand, the DS algorithm selects a subset of nodes such that every node which is not in the selected set, has at least one neighboring node which is selected. We apply the SELECTOR methodology to select a subset of algorithms from the full algorithm portfolio, using both meta-representations introduced above as graph nodes.

**Personalized problem portfolios.** Selecting algorithms based on the performance2vec and Shapley meta-representations allows us to find diverse algorithms based on overall performance (e.g., good, average, or poor performance). However, even selecting from a group of overall poorly

performing algorithms, theoretically, we can still select an algorithm that is the VBS for a particular problem instance, but not able to solve any of the other problem instances. To take into account algorithm performance complementarity on a problem level, we investigate *personalized* algorithm portfolios, with the main difference in the way we compute the meta-representations. To this end, we only consider the local Shapley meta-representations (i.e., per problem level) from performance prediction models. This allows us to create a separate graph for each benchmark problem, where the algorithms are nodes and edges connect them only if the similarity between their local Shapley meta-representations is greater than a predefined threshold. On each graph that represents a different benchmark problem, we run SELECTOR five times and return different sets for each problem class. We perform a union of the sets and then select the two top-performing algorithms from the union based on the raw performance data. Personalizing a portfolio can thus be seen as a combination of the data-driven and greedy approach.

## 3 Experimental Design

**Problem portfolio.** The problem portfolio consists of the first five instances of the 24 noiseless BBOB [HFRA09] problems in both $5D$ and $30D$, resulting in two sets of 120 problem instances each. As problem instance representations, we use 46 so-called "cheap" ELA features (which, in contrast to non-cheap ELA features, are based on non-adaptive sampling). Since our main focus is not on the benefit of automated algorithm selection *per se*, but the impact of the algorithm portfolio on its performance, we use feature values computed with comparatively large budgets, taken from [RDDD20] and available via the OPTION ontology [KVD+22a]. More precisely, we use the ELA features computed as the median of 100 independent repetitions of Sobol' sampling with $100D$ evaluations each. In all that follows, we ignore the cost for feature computation.

**Algorithm portfolio.** The algorithm portfolio consists of 324 configurations of the modular CMA-ES framework [dNVW+21]. We reuse performance data from [KVD+23], consisting of 10 independent repetitions on each problem instance. To measure algorithm performance, we consider a fixed-budget setting for budgets $B \in \{500, 2\,000, 5\,000, 10\,000, 50\,000\}$ of function evaluations (FEs), extracted using IOHanalyzer [WVY+22].

**Performance2vec meta-representation.** To generate the performance2vec meta-representations, we follow the procedure described in Section 2 and obtain, for each pair of problem dimension and budget, a 24-dimensional vector.

**Performance prediction models for Shapley meta-representation.** For each of the 324 modular CMA-ES variants, we train a single-target regression (STR) model, where for each problem instance represented by a vector of $p$ ELA features, we predict a real value $y$ that indicates the algorithm performance on that specific problem instance after a fixed budget. In line with previous works [KT19, CMMO13, JPED21] we opt for Random Forest (RF) regression models.

To tune the RF hyperparameters and evaluate the models performance, we use a two-stage nested cross-validation (CV) technique. The outer loop divides the data into training and test sets, while the inner loop determines the best hyperparameters for the model. In the outer loop, we perform a leave-one-instance-out (L1IO) CV, where one instance of each problem class belongs to the test set and all remaining four instances of each problem class belong to the training set; this way, we end up with five splits, corresponding to five instances of each problem. In the inner loop, we use the training splits that consist of the four instances from each problem class, 96 problem instances in total. We perform once again L1IO CV, where one instance from each problem class belongs to the validating set and three instances to the training set. This ends up with four splits. We then use a grid search to tune the RF hyperparameters on the training splits, and we select the best hyperparameters based on the average performance based on the $R^2$ score from the four validating datasets. After identifying the optimal hyperparameters for each modular CMA-ES configuration, we train the final model on all the outer-loop training data and evaluate it using the outer-loop test data. Since we have five splits, we repeat the training five times. For each

split, we compute the Shapley values that measure the contribution of each ELA feature to the end performance prediction for each problem instance from the test split. We then average the Shapley values across all problem instances from the test split, so we end up with five $p$-dimensional vectors for the same modular CMA-ES. Finally, to get the final meta-representation, we average each Shapley value across the five vectors arising from different test splits, yielding a more robust overall meta-representation. We have individual STR models for each combination of modular CMA-ES variant, problem dimension, and budget, resulting in a total of $324 \times 2 \times 5 = 3\,240$ trained models.

**Selecting diverse algorithm instances**. Using the meta-representations, we generate a graph with 324 nodes denoting the modular CMA-ES variants, as outlined in Section 2. We perform independent experiments for performance2vec-based and Shapley-based meta-representations. We consider the following similarity thresholds for adding an edge between two nodes: $\{0.60, 0.70, 0.80, 0.85, 0.90, 0.95, 0.97\}$. In total, we generate $2 \times 2 \times 5 \times 7 = 140$ graphs that correspond to different combinations of algorithm meta-representations (i.e., performance2vec or Shapley), two problem dimensionalities, five evaluation budgets, and seven different similarity thresholds. On each graph, we run the MIS and DS algorithms five times to select the diverse algorithm instances, due to the stochastic nature of the MIS algorithm.

**Personalized portfolios**. We need problem-level Shapley explanations to generate personalized portfolios. Since these explanations are available across five different testing splits, we average the Shapley values for each problem instance belonging to the same problem across all five splits. For each of the 24 BBOB problems, we construct a separate graph, where the 324 algorithms are connected based on their problem-level meta-representation similarity. On each graph, we run SELECTOR five times and return five sets for each problem class. We perform a union of the five sets and then select the two top-performing algorithm instances from the union based on the raw performance data. This is a combination of the data-driven and greedy approaches. We repeat this for each combination of problem dimension and budget, for two different similarities in constructing the graphs, 0.60 and 0.70.

**Building and evaluating the algorithm selector**. Our AAS makes the decision based on the output of regression models. The most suitable algorithm for each problem instance is defined as the one with the best-predicted performance value out of 324 regression models (i.e., one per each modular CMA-ES variant). We compare the AAS performance with a standard baseline: we select the virtual best solver (VBS), which is the true best algorithm for each problem instance. We evaluate the AAS by computing the difference (or *loss*) between the target precision of the selected algorithm $F_{\mathcal{A}}$ and the target precision of the per-instance VBS $F_{\mathcal{A}^*}$ for each instance as follows: $\mathcal{L}(\mathcal{A}, \mathcal{A}^*) = \log_{10}(F_{\mathcal{A}}) - \log_{10}(F_{\mathcal{A}^*})$.

We compare the loss distribution of the AAS w.r.t. the baseline-portfolio selectors in order to assess the AAS performance gains. To this end, we introduce three baseline algorithm portfolios: *(a) the full portfolio* which consists of all 324 modular CMA-ES variants, *(b) the greedy portfolio (auc)* where we select the top 10 configurations for each dimension based on anytime performance (i.e., the area under the empirical cumulative distribution function, which aggregates the fraction of performance targets reached across all runs at each given budget)[1], and *(c) the greedy portfolio (per-func)* where we select the top two best-performing configurations for each benchmark problem for each pair of problem dimension and a budget, ending up with 48 configurations (i.e., $24 \times 2$). To allow for a fair comparison between different portfolios, we select the VBS per problem instance out of the 324 algorithms, and that for all selectors generated with different algorithm portfolios (i.e., data-driven, full, greedy).

---

[1]The set of targets used here consists of 51 precision targets, logarithmically spaced between $10^2$ and $10^{-8}$, as is standard for the BBOB suite [HABT22].

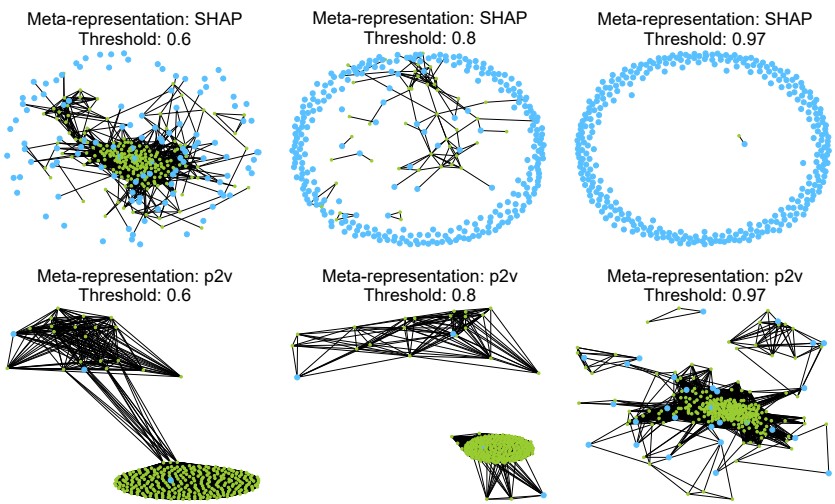

Figure 1: Graph representation of configurations selected with the MIS algorithm for problems of dimension 5, budget 2 000, and varying similarity thresholds. The nodes in blue are the nodes selected with the MIS algorithm while the nodes in green are not selected. The first row refers to graphs constructed with the Shapley meta-representation, while the second row refers to the performance2vec meta-representation. Each column refers to a different similarity threshold used to construct the graph (0.6, 0.8, or 0.97)

## 4 Results

We present our key findings; full results are available in our Zenodo repository [KCV+23].
**Exploratory data analysis of the selected algorithm portfolios.** Based on empirical evidence, portfolios yielded using the performance2vec meta-representations are of smaller size compared to the ones generated using the Shapley meta-representations (see Figure 5 in Appendix A). This comes as no surprise since the performance2vec meta-representations encode algorithm performance using high-level information and, as such, they are not able to ensure diversity like the one captured by the Shapley meta-representations. The latter encode the performance on a much lower level of granularity, taking into account landscape characteristics of problem instances. For further analysis, we select similarities of 0.80, 0.85, 0.90, 0.95, and 0.97 for performance2vec-based portfolios to ensure selecting more than two algorithms wherever possible (from 2 to 30). The problematic scenario is $30D$ with a cut-off budget of 500 FEs, where all 324 algorithms perform similarly, likely due to the fact that they are still in an exploration phase of their search procedure with such a small budget for optimizing problems in $30D$. In the case of Shapley-based portfolios, we select similarities of 0.60 and 0.70 in order to constrain the portfolio size as much as possible, but with resulting sizes ranging from 50 to 100 algorithms, they are still much larger than the performance2vec-based portfolios. Meta-representations with higher similarities suffer from over-capturing information, which yields an increase in the number of different algorithm behaviors. Both MIS and DS graph algorithms lead to a similar selection.

We take a closer look at the difference between the two meta-representations in Figure 1. We show the graphs constructed using different similarity thresholds for the Shapley and performance2vec meta-representations. We observe that, as the threshold value increases, the number of edges in the graph decreases; consequently, the number of nodes selected with the MIS algorithm increases. We also see that the performance2vec-based graphs are more strongly connected, meaning that fewer nodes are selected by the MIS algorithm compared to the graphs obtained using the Shapley meta-representation.

**AAS evaluation results**. To evaluate the overall performance of different selectors, we compute their total loss over the VBS $\mathcal{L}(\mathcal{A}, \mathcal{A}^*)$ and aggregate this over all instances of the 24 problems. Figure 2 visualizes average differences between the total loss of the AAS based on the full set of 324 algorithms and the total loss obtained by different portfolios selected by data-driven approaches. Since we apply the SELECTOR technique five independent times for both meta-representations, the depicted loss corresponds to the average over five portfolios generated from the same meta-representation.

Figure 2 contains results for two considered problem dimensionalities; the upper heatmap corresponds to $5D$, while the lower corresponds to $30D$. The rows in each heatmap represent different cut-off budgets, while the columns stand for the different approaches for selecting the algorithm portfolio. In the heatmap, values larger than zero (in blue) indicate that the selected portfolio performs better than the full portfolio, and vice versa for negative values (in red). We highlight the results of performance2vec-based portfolios with a similarity threshold of 0.95, and those with a similarity threshold of 0.70 for Shapley-based and personalized portfolios. These thresholds are chosen in such a way that the selected portfolios are of a reasonable size.

We notice several trends in Figure 2. For $5D$, the AAS based on personalized portfolios outperforms the one based on the full portfolio, but performs comparably with the greedy (auc)- and greedy (per-func)-based portfolios. For $30D$, the AAS based on personalized portfolios outperforms the AAS based on the full portfolio for all budgets, and also outperforms greedy (auc)- and greedy (per-func)-based portfolios for almost all budgets. On the other hand, the performance2vec-based portfolios seem to lead to worse total losses, and exhibit comparable performance to greedy (auc) and greedy (per-func) portfolios only in $5D$ for the budget of 500 FEs. In contrast, the Shapley-based portfolios yield better performance than the full portfolio for large budgets, where they are also comparable with both greedy approaches. Overall, the Shapley-based portfolios seem to slightly outperform the performance2vec-based ones. The former are more likely to contain overall good algorithms, whereas the latter are too sensitive and reactive to differences in algorithm performances. The distribution of the losses per portfolio and a fixed problem dimension, budget, and similarity threshold is available in our Zenodo repository (also see Figure 7 in Appendix C).

We now focus on personalized portfolios and give a more in-depth analysis of how portfolio selection affects the performance on a problem level. Figure 3 shows the distribution of the log-valued performance of the algorithm that is selected for $30D$ and 2 000 FEs per each BBOB problem separately. Each boxplot corresponds to log-valued target precisions for each problem instance belonging to problems 1 through 24; the blue stands for the VBS, the orange for the AAS on the full portfolio, the green for the AAS on the personalized portfolios (similarity of 0.70), and the red for the AAS on the greedy (auc)-based portfolio. We observe that all portfolios lead to results comparable with the VBS across most problems. These results are very promising when we consider the AAS task in a challenging setting of selecting among very similar solvers (all modular CMA-ES variants).

**AAS error decomposition**. When selecting a smaller algorithm portfolio from a large set of algorithm instances, there are two main aspects to consider. The first is the inherent limitations of the selected portfolio: if the best configuration on a certain problem instance is not included, the algorithm selector can never match the original, full-portfolio-based VBS. The second factor is the difficulty of the actual AAS task: if a portfolio contains more algorithm instances, the chance to make incorrect choices increases.

To analyze the contribution of these two aspects, we decompose the total loss of our algorithm selector (relative to the original VBS) and visualize this trade-off for a set of portfolios constructed by each of the methods mentioned in Section 2. Figure 4 shows this error decomposition for $5D$ and a budget of 2 000 FEs. Our goal is to find portfolios that have *inner loss ≤ greedy - outer loss* and also outperform the full portfolio. The full portfolio is represented as the diagonal that goes through 2 on the $x$- and $y$-axis. The area of interest is on the left and below this diagonal. Each

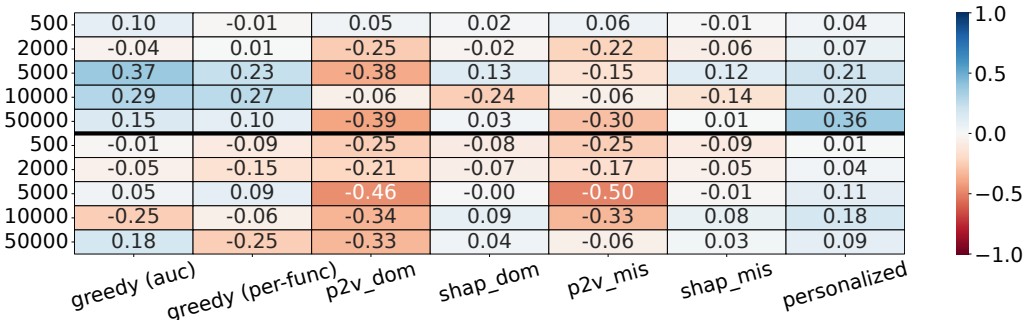

Figure 2: Difference between the total loss of the AAS trained on the full algorithm portfolio and the AAS trained and selecting only from a small selected portfolio (positive values in blue = AAS for subset is better). Top: 5*D*, bottom: 30*D*, rows = different evaluation budgets. Note that loss values are based on differences in logarithmic precision, which is aggregated across all functions.

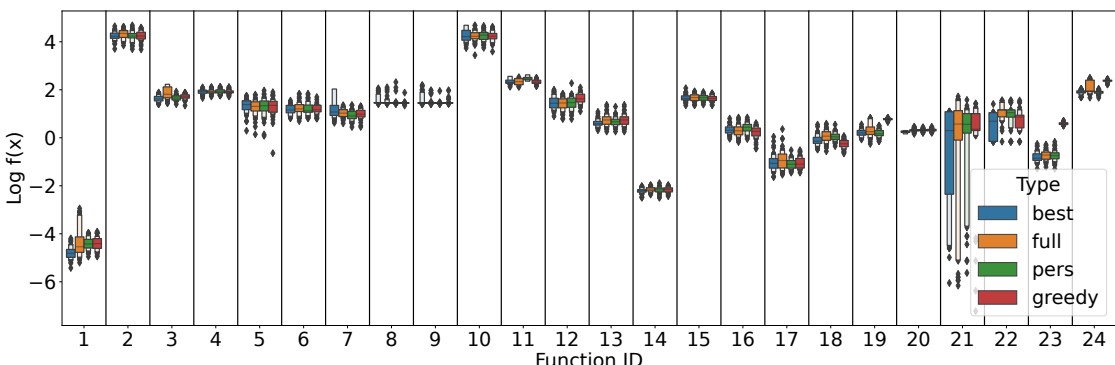

Figure 3: The distribution of the logarithmic value of the precision for the algorithm configuration that has been selected for each 30*D* instance (budget 2 000), aggregated on a per-problem level. 'Best' corresponds to the VBS, 'full, 'pers' and 'greedy' refer to the ASSs based on the full, the personalized (similarity 0.7) and greedy AUC-based portfolios respectively.

marker in the figure corresponds to the inner and outer losses of the AAS built with different portfolios. Every point for performance2vec-based and Shapley-based portfolio stands for the loss obtained across different runs (i.e., seeds) of SELECTOR. From this figure, we see that several performance2vec-based portfolios lead to the lowest total loss (no matter that they were overall the worst across all seeds). Similar observations can be drawn for the Shapley-based portfolios. Both personalized portfolios constructed only with the Shapley meta-representations yield smaller total losses than the greedy (auc)- and the greedy (per-fun)-based ones. Similar results are also presented for other combinations of problem dimensionality and budgets (i.e., (5*D*, 500 FEs), (5*D*, 50 000 FEs)). In the case of 5*D* and 5 000 functions evaluation, there are two Shapley portfolios better than the greedy (auc). For 5*D* and 10 000 function evaluations, one personalized portfolio is close to the greedy (auc) and the greedy (per-fun). For 30*D*, budgets of 500 and 50 000 function evaluations lead to a few portfolios that have lower total loss than the greedy (auc). While 2 000, 5 000, and 10 000 function evaluation leads to a large number of portfolios that perform better than the greedy (auc). The minimization of the total loss can be also viewed as a bi-objective *min-min* optimization problem. In all pairs of problem dimensions and budgets, this analysis also provides

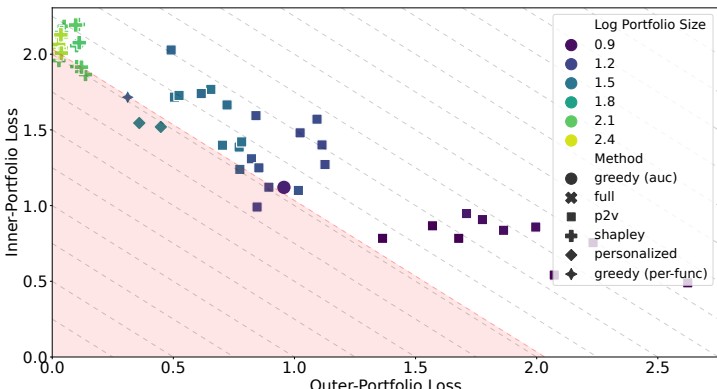

Figure 4: Error decomposition for the selected portfolios, for 5$D$ with budget 2 000. The x-axis is the best possible loss of the portfolio, so the difference between the portfolio's VBS and the VBS of the full set of 324 algorithms. The y-axis is the loss of the algorithm selection, so the difference in performance between the algorithm it selects and the VBS of the portfolio it can choose from. Both of these losses are averages across all instances.

an approximation of the Pareto front which consists of our selected portfolios and which shows that they are competitive with the greedy (auc)-based portfolio.

## 5  Discussion

In this work, we have investigated the trade-off between the flexibility to choose from a large algorithm portfolio and the increased complexity of the selection task that larger portfolios exhibit. We observe that the approach based on the performance2vec meta-representation provides smaller portfolios, which do not necessarily contain the best configuration for each problem instance, but which substantially gain from a much improved performance of the selector trained and operating on this small portfolio. In contrast, portfolios selected based on Shapley meta-representations are larger, more often comprising the best algorithm for a given problem instance, but at the cost of the trained AAS not always selecting it. The strength of the Shapley meta-representations is that the selected portfolios provide results as competitive as a greedy portfolio does. This suggests that the step of selecting a portfolio of diverse and complementary algorithms can be automated and there is untapped potential in the interaction of algorithms and problem instances.

The main drawback of the investigated approaches lies in the computational cost, especially that for obtaining the Shapley meta-representations since they require training the regression models for each algorithm from the full portfolio and analyzing them with the explainable SHAP post-hoc analysis. However, since a lot of explainable automated algorithm performance prediction analyses [KVD$^+$22b, NLKE22] have been already performed and offer publicly available results, in the future we will focus on making such results interoperable across studies by including them in semantic models such as OPTION [KVD$^+$22a] and further reusing these results for a selection of portfolios. This will require training the regression models only for a small portion of algorithms to learn their meta-representations (i.e., those not included in the semantic model), while the meta-representations of other algorithms will be available from previous studies. We point out that Shapley meta-representations are model-specific (in our case obtained from RFs), which suggests similar analyses as above can be repeated for other regression models such as XGBoost, Deep Neural Networks, etc.

The personalized portfolios selected based on the Shapley meta-representations and using a greedy post-hoc phase yield better results than the purely greedy approaches. These portfolios also

allow more flexibility to include those algorithms that are not in the top best-performing ones based on the ranking, but they exhibit only small differences in performance, which are not statically significant for the results obtained by the top best-performing algorithms.

The design of novel meta-representations of black-box optimization algorithms has not been within the scope of this paper. It is a complex task that merits its own research. The designed meta-representations must be evaluated in detail and demonstrated to work well before being used in downstream applications such as ours. For this reason, we have chosen to rely on previously proposed meta-representations and demonstrate how they can be used to select complementary algorithms for an algorithm portfolio. However, our suggested approach is general and can be applied using any meta-representations developed in the future.

Although the main focus of this work is on practitioners who specialize in black-box optimization, it should be noted that our AAS study is analogous to hyper-parameter optimization (HPO) as it involves choosing among different variants of the CMA-ES to compose a portfolio. This paper is therefore relevant to both the black-box optimization and the machine learning communities, and the problem introduced here is an AutoML problem at its core. The findings from this study are shedding new light on the impact of algorithm portfolio selection on AAS performance, which is significant for the field of AAS from both perspectives.

## 6 Conclusion and Future Work

Automated algorithm selection (AAS) performance is highly dependent on the algorithm portfolio. Choosing a portfolio is difficult, as it involves balancing the flexibility of the portfolio with the increasing complexity of AAS. Typically, algorithms are chosen based on performance on reference tasks. In this paper, we have explored data-driven selection techniques for building a portfolio of algorithms. Our method uses algorithm behavior meta-representations to create a graph and select a diverse and representative algorithm portfolio. We evaluated two meta-representation techniques (SHAP and performance2vec) for selecting portfolios from 324 CMA-ES variants for BBOB single-objective problems. Two types of portfolios were investigated: related to the overall algorithm behavior and personalized related to algorithm behavior per each problem. Performance2vec-based portfolios favor small sizes with minimal AAS error relative to the VBS from the selected portfolio, while SHAP-based portfolios are more flexible but have lower AAS performance. Personalized portfolios provide comparable or slightly better results in almost all cases the classical greedy approach of selecting best-performing algorithms based on reference tasks. In addition, they outperform the full portfolio in all cases.

Future research avenues include testing the methodology for other modular frameworks such as modular DE [VCKB23] and modular PSO [BWB20], as well as examining the possible use for more inherently diverse portfolios. Another further direction is designing novel meta-representations for describing algorithm behavior. To this end, it is worthwhile investigating trajectory-based meta-representations which will be able to capture the internal dynamics of an algorithm applied to a particular problem instance during the optimization process. We can learn these meta-representations based on candidate solutions that are observed by the algorithm running on that particular problem instance. Last but not least, we plan to test the selected portfolios on other benchmark suites, to estimate if a data-driven algorithm portfolio trained on one benchmark suite is able to generalize well to other suites.

**Acknowledgements**. The authors acknowledge the support of the Slovenian Research Agency through program grant No. P2-0098 and P2-0103, project grants N2-0239 and J2-4460, young research grant No. PR-12393 and No. PR-09773, a bilateral project between Slovenia and France grant No. BI-FR/23-24-PROTEUS-001 (PR-12040), as well as the EC through grant No. 952215 (TAILOR). Our work is also supported by ANR-22-ERCS-0003-01 project VARIATION.

**Mandatory impact statement**: Our work classifies as fundamental research with no noticeable negative impact on the society or the environment.

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

# A  Sizes of the Selected Algorithm Portfolios

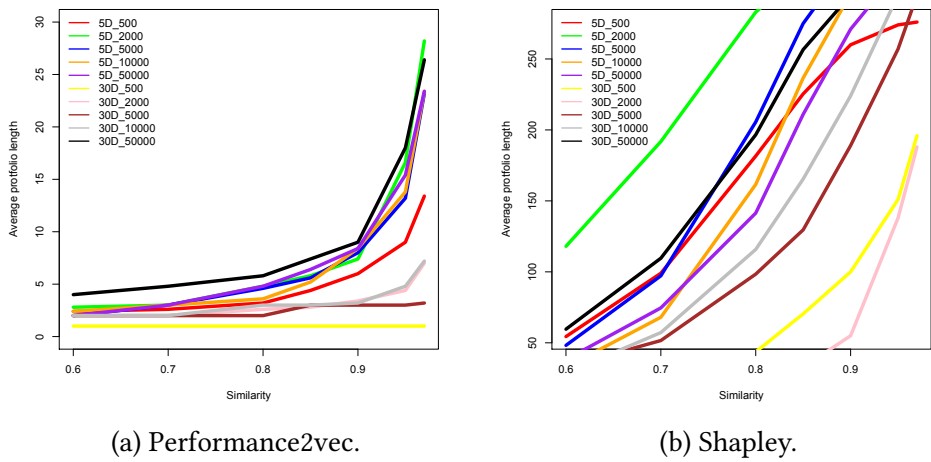

(a) Performance2vec.     (b) Shapley.

Figure 5: The average portfolio size obtained across five runs of SELECTOR for each triplet of problem dimension, budget, and similarity threshold for generating the graph, separately for each algorithm-behavior meta-representations.

# B  Benchmarking with Randomly Selected Portfolios

To investigate and benchmark the AAS results when random portfolios are used, we fixed the problem dimension on $5D$ and the budget of 2000 function evaluations. Then, for both performance2vec (0.95) and Shapley (0.70) meta-representations and a single run of SELECTOR, we select three random portfolios with the sample size returned by SELECTOR. In the case of performance2vec, we randomly select 16 algorithm instances, while in the case of Shapley 187 algorithm instances. Figure 6 presents the AAS losses calculated with regard to the VBS selected out of 324 configurations. From it, we can see that in the case of both meta-representations, the selected portfolios provide more robust results than the randomly selected portfolios with the same sample size.

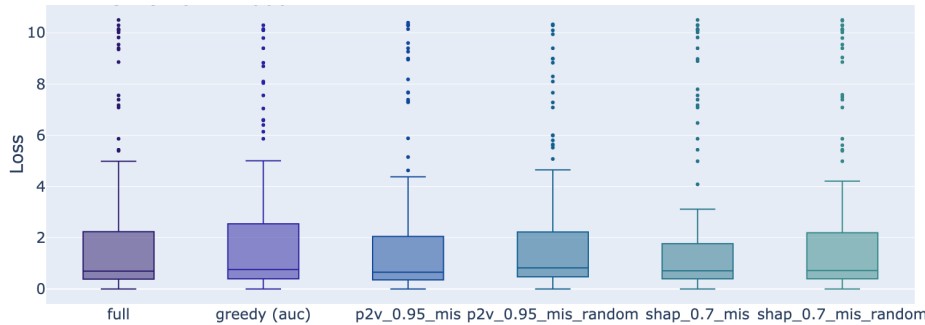

Figure 6: The distribution losses of the AAS with different portfolios across 120 $5D$-problem instances with regard to the VBS selected from the full portfolio of 324 configurations. Benchmarking a fixed portfolio returned by a single run of SELECTOR for both meta-representations (p2v and Shapley), with a corresponding randomly selected portfolio with the same sample size (16 for p2v and 187 for Shapley).

## C Distribution Losses of the AAS with Different Portfolios

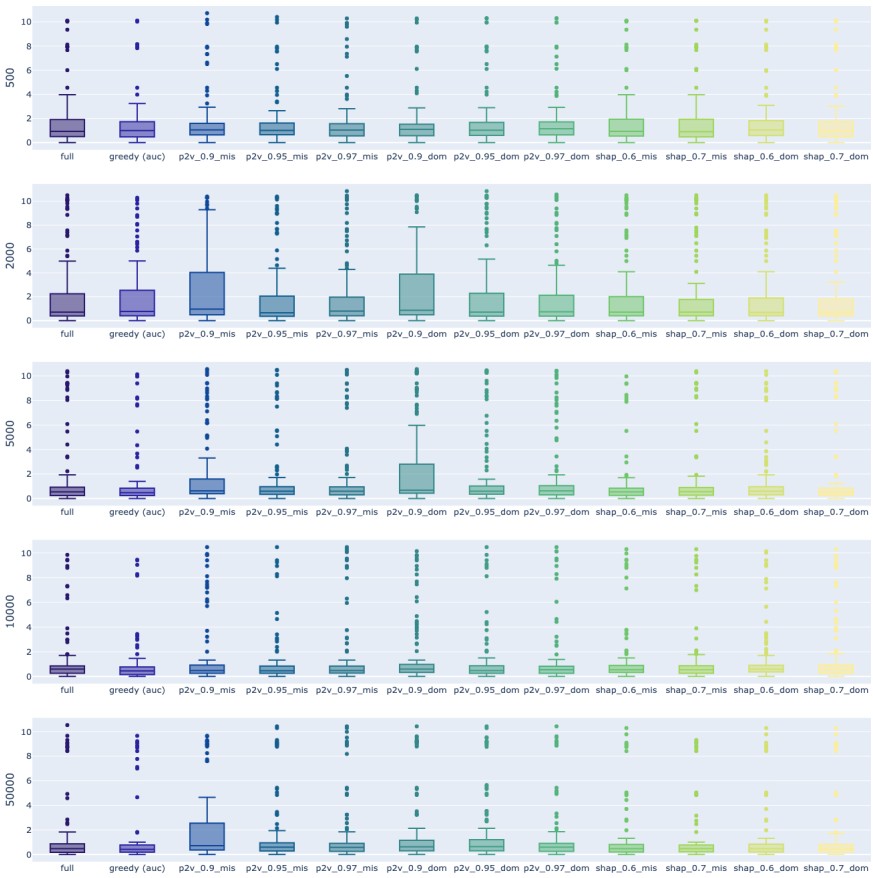

Figure 7: The distribution losses of the AAS with different portfolios across 120 5*D*-problem instances with regard to the VBS selected from the full portfolio of 324 configurations. Since all portfolios are generated by SELECTOR using performance2vec (p2v) and Shapley (shap) meta-representations using five runs, we aggregated the losses per-problem instance level using the median loss of the five runs.

## D Data reuse

For this paper, we make use of existing data which has been made publicly available. In particular, the performance data has been taken from [KVD+23], licensed with Creative Commons Attribution 4.0 International license. The ELA data has been taken from [RDDD20], also with a CC4.0 license.

