# OpenReview forum: "PS-AAS: Portfolio Selection for Automated Algorithm Selection in Black-Box Optimization"
_automl.cc/AutoML/2023/Conference — AutoML 2023 MainTrack_

### Review · Reproducibility_Reviewer_9dcF · 2023-04-10

**Completeness Of Code And Dataset Supplement Rating:** 4
**Usability And Ease Of Reproducibility Rating:** 2

**Actions Required To Increase The Reproducibility And Overall Recommendation:**

I will raise my score if the authors resolve the minor comments I have made about the checklist and the missing components of the code.
Additionally, the authors should provide more detailed documentation (for each component/folder).

To raise my score even further, I would like to see a step-by-step guide on how to use all the components of the work to reproduce the results. E.g.: first, install x, then run part 1 file 1, then file 2, then file 3 with these specific arguments. This can also be done per folder.
Importantly, the step-by-step guide should work such that I only need to download the code and data and follow it without having to adjust the file paths in the script files manually.

The point that I raised related to "Information Leakage during the Evaluation of the Algorithm Selection" does not affect my rating, since it is not directly related to reproducibility. Nevertheless, I think it might be important for the meta-reviewer to have this clarified.

**Completeness Of Code And Dataset Supplement:**

The authors have provided code and dataset supplements for almost all of the points mentioned above.

I only found the following minor points to be missing:
* I have not found the code for the Heatmap (Figure 2) or the Boxplot per BBOB function (Figure 3). But, I have found the figure files for both.
* There seems to be some code missing for minor preprocessing between different folders. For example, transforming the data collected and stored in the "Performance_Data" folder to what is used in the "Algorithm performance prediction/raw_data/Performance_tables_modcma". Besides this case, I found this to be the case for the `conf_perf2vec.csv` in the `SELECTOR` folder. However, as far as I can tell from comparing the files before and after the transformation, the content is identical in both cases.



**Overall Reproducibility Review:**

## Positives

The authors provided all data they created and used. This allows other researchers to re-use the data easily.

The authors provided almost all of the code to reproduce the results and figures in the paper; allowing others to use the code and inspect the implementation.

For all parts that I could test, the code can reproduce the results of the paper. For the one part that I could not run myself but for which I analyzed the source code, I believe it to be highly likely that it would reproduce the results as shown in the paper.

The checklist is mainly filled reasonably. Thus, allowing others to understand important details of the work quickly.

### Negatives

The documentation and code quality of the provided code can be improved a lot and make it hard to use the code or to understand it.

The provided code includes several aspects that are not part of the final paper and are out of place for this submission. This makes the code more confusing than it has to be.

It is unclear which code/data is from the authors or previous work with the current documentation. As a result, the exact contribution of the paper is hard to determine, and, in its current state, the repository re-uses code from other researchers without referencing them.



**Review Confidence:**

4: You are confident in your assessment, but not absolutely certain. It is unlikely, but not impossible, that you did not understand some parts of the submission or that you are unfamiliar with some pieces of the code or data.

**Review Rating:**

8: Accept, all aspects of this are reproducible with minor effort.

**Review Summary:**

The authors provided all necessary components to make their code reproducible but failed to provide the necessary documentation and additional code to make it reproducible without major effort. Other research would have difficulty using or building upon the work using the code. Moreover, the missing documentation and current project structure make it unnecessarily complicated to comprehend the approaches presented in the paper.

**Summary Of Necessary Code And Dataset Supplement:**

The authors' description and statements in the paper require the following code or dataset supplements:

Data Creation
- code for reusing performance data of 324 modular CMA-ES variants with data taken from previous work
	-- code to extract the five different fixed-budget settings using IOHanalyzer from the performance data
	-- performance data for applying CMA-ES variants to 24 BBOB problems (first five instances)
- code for reusing data of 46 "cheap" ELA features with data taken from previous work available via OPTION ontology
- code for performance predictions (single target random forest (RF) regression model for each CMA-ES variant)
	-- code for model evaluation using leave-one-instance-out cross-validation (outer split)
	-- code for RF hyperparameter tuning with grid search using leave-one-instance-out cross-validation (inner split, optimizing for average R^2 score)
	-- code to re-train the RF model after tuning

Portfolio Creation
- code for performance2vec and SHAP meta-representations of algorithm behavior

- code for running the SELECTOR methodology (five times MIS and DS algorithm) on graphs produced from the meta-representations of performance2vec, Shapely, and local Shapley;
	-- code to generate graphs from the meta-representations (for local Shaple,y do this only for two similarity thresholds with the local shapely explanations based on average across the testing splits of the leave-one-instance-out cross-validation (outer split))
- code to create two types of portfolios (focusing on overall behavior vs. personalized to each problem's behavior)
- code for random selection of portfolio members and its evaluation (Appendix B)

Algorithm Selection
- code for ASS at five different cut-off budgets (500, 2k, 5k, 10k, or 50k function evaluations) with two problem dimensions (5D and 30D)
- code to compute the VBS(s) and the three baseline algorithm portfolios
- code to generate the plots from raw result data


The code and dataset supplements were shared via https://zenodo.org/record/7763969 (last updated by authors on March 22, 2023; last accessed and downloaded by me on April 10, 2023).

**Usability And Ease Of Reproducibility:**

# Summary
While I could replicate most of the results presented in the paper, the usability of the code was problematic.
The supplements require a better project structure and more detailed documentation to make this easy to reproduce (in the sense of replicating). Moreover, I believe that the current state hinders other research from using and building upon the code. In essence, a step-by-step guide to run all parts of the code (e.g., from the CLI) after downloading the repository would help a lot.

*The following details all problems I have encountered and deem necessary enough to raise here.
Moreover, in the end, I am raising a point related to the implemented evaluation that I deem very important.*


## Comments for the authors on "Usability And Ease Of Reproducibility"
Most of these points address the authors and are less relevant to the review. **However, critical points are highlighted with bold text**.
The comments are split based on the parts as divided in the code/readme, general, and other.

### General
* To my understanding, the authors are re-using data and code from previous work at several points. This has been mentioned in the paper (e.g., Line 127). But, the code and documentation do not clarify what parts of the code are contributions of the authors or previous work. This becomes confusing for the re-used ELA and performance data, as there are no references to other work in the repository. For instance, the `Performance_Data/data_collection.py` seems to be from previous work, but I cannot tell for sure.

### Part 1: Performance data collection (Folder: `Performance_Data`)

* The `requirements.txt` is confusing and states requirements for the R code (but R does not support a `requirements.txt` file for installation). Moreover, the Python scripts require `ioh` and `modcma`(as mentioned in the readme), but the needed version for neither is given.
* In `data_collection.py`, I had to replace `REAL` with `ioh.ProblemClass.REAL` for `ioh.Experiment` to make it run.
* Running `data_collection.py` does not replicate its results stored in the repository. This seems to be an artifact of randomness as it is not controlled by random states or similar (nor does this seem to be supported by the tools used). This might require some documentation explaining why this happens.
* The paths in `Data_processing.R` are not correctly configured for the project structure after downloading the repository (even after renaming the folders as instructed in the readme); consider using relative paths or allow the base path to be configurable.
* The code works with one more budget (20k) than described in the paper.

### Part 2: Algorithm performance prediction (Folder: `Algorithm performance prediction`)

* The `requirements.txt` is missing `shap` and its version.
* There seems to be no reason these files are notebooks, as the notebooks do not offer additional documentation. But introduce the overhead of using a notebook client like Jupyter. I converted them to script files during my usage.
* The paths in the Python files are broken and do not work out of the box.
* The notebooks contain code for modular Differential Evolution and code for testing different meta-learning algorithms besides RF (XT, MLP, XGBoost). I have ignored this code as its usage is not mentioned in the paper.
* The documentation on how to use `2-train_predict_shapley_RForest.py` is missing, especially what it requires as input arguments (configuration IDs in what range?).
* Some comments and paths in `2-train_predict_shapley_RForest.py` should be removed or anonymized.
* **The code evaluates scaling and not scaling of the input data as part of the grid search for RF, but this was never mentioned in the paper. This prevails throughout the rest of the code, and the results with scaling are used multiple times. However, it is unclear whether the final meta-representation was achieved with scaling or not. My best guess is that the final shapley portfolio is created without rescaling (see `generated_portfolios.R` code).** Importantly, unlike the hyperparameters of RF, scaling is not part of the selection mechanism used during grid search.
* If scaling was used in the end, then this is problematic because **scaling was implemented incorrectly**. The sklearn-based scalers are fit on the test data for transforming the test data. But, they should only be fitted on the train data -- **`scaler.fit(X_test_inner)` as well as `scaler.fit(X_test_outer)` in `2-train_predict_shapley_RForest.py` must be removed to scale the data without information leakage correctly or misrepresented scaling**, see its [documentation](https://scikit-learn.org/stable/modules/preprocessing.html#preprocessing-scaler).

### Part 3: SELECTOR (Folder: `SELECTOR`)
* I could not install the requirements needed for the SELECTOR code. The requirements file contains 170 dependencies, all of which are version locked. Python's pip failed to resolve this such that all requirements could be satisfied. Moreover, the required Python version and used OS is missing. Hence, I only analyzed the source code for this section. Furthermore, it is again unclear if this code was created by the authors or taken from previous work (as mentioned in the paper).
* Details on the usage of the files in this folder are missing, Specifically, what each file does and how to use them (e.g., `dom_mis_run_5_times.sh`?).
* Here, the notebook is appropriate since it draws the figures. However, it contains some random code for heatmaps at the end. Additionally, it includes some analyses not reported in the paper.


### Part 4: Selecting data-driven portfolios (Folder: `PORTFOLIOS` and `Portfolio`)
* There are two directories for this part for some reason. One contains only the data; the other data and the code.
* As a minor comment, the readme section for Part 4 has several typos.

### Part 5: Algorithm selector (Folder: `Algorithm selector folder`)
* I again do not see why one would use a notebook-based code. Nothing is printed, nor any plots show in the notebook. Moreover, the notebook is enormous. A simple script-file-based project structure would improve usability immensely. Finally, the notebook has many code repetitions that could be abstracted into re-used functions.

### Part 6: Error decomposition (Folder: `Error_Decomposition`)
* The file paths for part 6 are not working for the project structure after the download.

### Other
The method used to share the code and dataset supplements had a minor problem I would like to raise: the authors shared the data and code together. As Zenodo had a strangely behaving download speed varying between 600 k/bs and three mb/s, it took me 4 hours to download all data. Perhaps it would be better to share code and data separately in the future (using something like anonymous GitHub for the code and Zenodo for the data). Thus, making it easier for other researchers to directly access the code such that they only have to download the data when needed.

In Line 162 of the paper, the authors mention that their approach resulted in `324 x 2 x 5 = 3240` trained models. But, as far as I can tell from the code and the description in the paper, it results in `324 x 2 x 5 x 5 = 16200` models, including one for each of the five folds of cross-validation. The predictions of all these models are used for algorithm selection and sharply-based meta-representation.


## Information Leakage during the Evaluation of Algorithm Selection?
After reading the paper and going through the code, I want the authors to clarify a specific aspect of their evaluation related to the shapely meta-representation created using cross-validation.
In its current state, I believe -- with medium confidence -- that **the algorithm selection might be affected by information leakage**.
I would like to understand the evaluation correctly and clear up my doubts.

To explain, let me first recap my understanding of how the final algorithm selection is happening with the shapely meta-representation:

1) Given the performance and ELA data for all CMA variants and the problem instances, we compute the shapely meta-representation by producing shapley explanations for a fold's *test data* during cross-validation, i.e., explanations for the Out-Of-Fold (OOF) predictions of a tuned RF (trying to predict the performance data given ELA data for all CMA variants). At the same time, we compute the OOF predictions and store them for later.
2) Next, we create one dataset from the explanations for each fold; let's call this `test_data_explanations`. Then, we use the SELECTOR approach on a graph built from the `test_data_explanations` to generate a portfolio $P_{test}$.
3) Finally, we evaluate the algorithm selection. Therefore, we collect all OOF predictions for each variant in $P_{test}$. We then select the variant with the smallest predicted loss for an instance -- the smallest value from the OOF predictions for this instance.

There seems to be information leakage because we evaluate the algorithm selection in a cross-validation-like setting (by using the OOF predictions), but at the same time, the $P_{test}$ was created using `test_data_explanations`. `test_data_explanations` contains data for test data from other folds, which is training data in the current fold, that should not be available for selecting the best among the OOF predictions for an instance of the present fold.
This might be even more server for the personalized (local shapely) approach as we directly aggregate across folds (instances) for the final meta-representation.
*In my understanding, one should only build the meta-representation per fold and not one representation across all folds.*

---

> ### Author Response · Authors · 2023-05-01
> **Response to Reproducibility Reviewer 9dcF**
>
> We thank you for your helpful and valuable comments, especially making our code more easy to be executed. Some of the reproducibility comments will be fixed when the authors are back from holidays (tentatively until May 7th) since it is a spring holiday season for most of the team members.
>
> ### Reproducibility ###
>
> General
> We have updated the readme to more clearly indicate which code and data are taken from previous work.
>
> Part 1
> The requirements file has been modified to now be relevant for the Python part of this code. For the R-specific requirements, the version number has been added to the readme and a link provided to the github page (where more detailed installation instructions can be found).
> The problem with the REAL-argument for IOH was related to a recent update in the package. We have updated the code accordingly. We also added an explicit reference to the paper from which we take this code and data.  Unfortunately, the handling of the seeds was not done originally, so re-running this script gives slightly different results, although the amount of runs done should still be sufficient for drawing statistical conclusions.
> Finally, the R-script has been updated to properly use relative paths to a user-specified working directory, and this has been updated in the readme. For usability and easier downloading, the data collection files and the performance data have been split into two separate .zip files.
>
> Part 2
> With regard to the scaling and not scaling experiments, we should point out that the experiments in the study are performed only by using the non-scaling variant. This is why this is not explicitly mentioned in the paper. The scaling experiments have been performed only to check if there are differences in the ML performances. We will clean this up after the holidays within the repo and make it clear.
>
> Part 3
> The requirements file has been refactored to enable easier installation of the required libraries.
> Python 3.7 or 3.8 is recommended to be used. The experiments were run on Ubuntu 21.10 OS.
> The notebook has been refactored to remove the analysis not reported in the paper.
> The dom_mis_run_5_times.sh file enables running the experiments in a centralized manner. It calls the python scripts which run the SELECTOR methodology with the SHAP, Performance2Vec and the personalized meta-representations. We added this as a more detailed guide.
>
> Part 4
> The typos in the description have been fixed. We have two folders that contain the same generated portfolios. We will restructure this and connect the PORTFOLIOS folder to the Error decomposition experiment.
>
> Part 5
> We will fix this after the holidays and make the code available as a script and not a notebook.
>
> Part 6
> The notebook has been updated to better deal with the project's folder structure. It now uses relative paths which by default are set to the names as in the Zenodo repository. In addition, the notebook has been extended to include the code for generating figures 2 and 3 from the paper, which were missing before.
>
>
> ### Specific comments (Submission Checklist)
> * 1b) The answer is  "Yes". It was our previous answer and also the suggestion from the reviewer.
> * 3a) We will work on the “execution commands” after the holidays. Thank you for the detailed comments, we improved the code and its documentation a lot.
> * 3c) The code for all figures was added.
> * 3d) Thank you for your detailed comments and constructive feedback. Some of the above-mentioned comments have been already fixed. However, the remainder will be fixed after the holidays. We believe that after resolving the above-mentioned issues the answer will be  “Yes”.
> * 3e) Sorry for making a confusion. The scaling experiment has not been reported in this study. It has been performed only for our sensitivity analysis, so we cleaned up the data available in our repository. Re: hyperparameters, we reported in the paper how we selected them. Due to the page limit, we did not include details in the paper. After the holidays, we will make a table of selected hyper-parameters available in our repository.
> * 3j) We mention that the seeds are based on the run_id.
> * 3n) We clarified this based on your comment. We set this to “No”, we explained the hyper-parameter tuning procedure without the time spent to perform it.
> * 4b) This does not mention the licenses of the used assets (e.g., from the other papers) and might be more appropriate to state in the Appendix and reference it from the checklist (same for 4a).

---

> > ### Author Response · Authors · 2023-05-01
> > **Response (continued): information leakage**
> >
> > Thank you for the concern and your explanation. We agree that there can be information leakage only in the case of personalized portfolios. This is due to the fact that we do not have separate test data, due to the relatively low overall number of problem instances we focus on (unlike in typical ML applications, where datasets are much larger). However, we believe that this information leakage does not have a crucial impact in the AS performance if we take into account the way our training and testing data sets are constructed. Our stratified cross-validation guarantees that problem instances of each problem class which differ in shifting and scaling are present both in the training and testing data. We point out that our study involves only the most commonly used benchmark suite for black-box optimization, however in future we plan to use other recently published benchmark suites that contain problem generators and have better coverage on the problem space. This will allow us to have fully separate test datasets.

---

> > ### Comment · Reproducibility_Reviewer_9dcF · 2023-05-02
> > **Response: Changes for Reproducibility**
> >
> > Dear Authors,
> >
> > Thank you for the changes and the major additional effort put into making your work reproducible. I am convinced from the response that you have adequately updated (or will update) the code to make their work easily reproducibly. I have raised my score accordingly.
> >
> >
> > Note, I have not been able to verify this personally as I have not found the update code on zenodo.
> > Moreover, I likely won't have the time before the rebuttal ends. Still, I am happy to verify this during the discussion period if the area chair wants me to (once all changes were added as mentioned by the reviewers).

---

> > > ### Author Response · Authors · 2023-05-02
> > > **Response: New DOI**
> > >
> > > Dear Reproducibility Reviewer,
> > >
> > > We warmly thank you for updating the score according to the changes we provided during the rebuttal phase.
> > > Please note that there is a new DOI of the Zenodo repo in the revised version of the paper: https://doi.org/10.5281/zenodo.7866416 containing changes implemented so far, as well as a completely new description of the repo (that, for the time being, serves as a README file, until we fully implement the remainder of the modifications).
> > >
> > > We apologize for the confusion this might have caused!

---

### Official Review · Reviewer_BXUD · 2023-04-12

**Potential Impact On The Field Of Automl Rating:** 2
**Technical Quality And Correctness:** The experiments seem well-done. I hav…
**Technical Quality And Correctness Rating:** 4
**Clarity Rating:** 3

**Summary Of Contributions:**

The paper investigates "data-driven" approach for portfolio selection in the context of black-box optimization. The authors evaluated two ways to characterize algorithms -- performance2vec and Shapley based characterization. The paper also explores the idea of "personalized" portfolio building where a portfolio is built for each problem type. The experiments show that using algorithm meta-representation to build portfolios doesn't usually help with algorithm selection. However personalized portfolios yield the best performance.

**Actions Required To Increase Overall Recommendation:**

I think the paper would benefit from better motivation for the work. For example, you mention that you used 324 algorithms in your study and the introduction allures to the problem of large number of algorithms in full portfolio. Maybe mention the number of algorithms in the beginning? I think that can motivate the paper better.
I'm not convinced that Shapley values can serve as meta-representation of algorithms. Can you provide your reasoning why you chose to use Shapley values?

**Clarity:**

Overall, the paper is well-written. However, can you add more details regarding Maximal Independent Sets (MIS) and Dominating Sets (DS)?

**Overall Review:**

Negative:
- mostly exploratory work
- I'm not convinced that Shapley values can serve as meta-representation of algorithms
- No clear motivation (why is this a problem to begin with?)

Positive:
- Nice experimental setup


**Potential Impact On The Field Of Automl:**

The paper shows a lot of exploratory experiments on portfolio building in the context of black-box optimization. But I'm not sure what the main message of the paper is. I think it's a well-done exploratory work but I'm not sure what its contributions are to AutoML community.

**Review Confidence:**

3: You are fairly confident in your assessment. It is possible that you did not understand some parts of the submission or that you are unfamiliar with some pieces of related work.

**Review Rating:**

4: Weak Reject: For instance, a paper with minor technical flaws, limited impact, and/or weak evaluation.

**Review Summary:**

Overall, I think the paper is a nice exploratory work of portfolio selection but there are a couple of comments I have about the paper (see in other sections). I'm not convinced that it has a large impact in the AutoML.

---

> ### Author Response · Authors · 2023-05-01
> **Response to Reviewer BXUD**
>
> We thank you for your helpful and valuable comments. In the following, we address your individual comments. Please refer to the newly updated version of the paper for modifications.
>
> ### Better motivation ###
>
> We made sure to better motivate the problem by mentioning the importance of the algorithm selection (AS) task when we deal with hundreds of possible algorithms. However, we strongly believe that investigating the trade-off between the flexibility of the portfolio and the complexity of the AS task is a research question that merits zooming into, what we have attempted to do with this work. Understanding these relations could further help practitioners to position their use-cases and employ the most adequate strategy, whether it is having more complementary algorithms in the portfolio to begin with, or reducing the AS complexity to a minimum.
>
> ### Clarity ###
>
> We added more details regarding Maximal Independent Sets (MIS) and Dominating Sets (DS).
>
> ### Shapley values ###
>
> Shapley values quantify individual contribution of input features to the overall performance of the ML model. These contributions capture essential information about what can be thought of algorithm-instance interaction, both on a local and a global level. For these reasons, we believe that the SHAP method can be effectively used as a possible algorithm meta-representation, despite being instance-specific.

---

### Official Review · Reviewer_C1SQ · 2023-04-13

**Potential Impact On The Field Of Automl Rating:** 3
**Technical Quality And Correctness Rating:** 3
**Clarity Rating:** 3

**Summary Of Contributions:**

It is well known that the effectiveness of automated algorithm selection heavily relies on the selection of a suitable algorithm portfolio. This task involves finding a balance between the flexibility of a large portfolio and the increased complexity of the AAS process. Typically, algorithms for the portfolio are chosen using a greedy selection approach based on their performance in some reference tasks.

The purpose of the paper is to explore alternative, data-driven techniques for algorithm portfolio selection. Tha authors introduce a method that involves learning meta-representations of algorithm behavior, constructing a graph based on the similarity of these representations, and applying a graph algorithm to select a final portfolio of diverse and non-redundant algorithms.

The effectiveness of two distinct meta-representation techniques is assessed by selecting complementary portfolios from a set of 324 CMA-ES algorithm variants for optimizing classic BBOB problems. The experimental results show that the performance based on performance2vec representations tend to favor small portfolios with negligible error in the AAS task, while the SHAP-based representations offer more flexibility but decreased AAS performance. Personalized portfolios perform comparably or slightly better than the greedy approach and outperform the full portfolio in all scenarios.


**Actions Required To Increase Overall Recommendation:**

The impact is limited. The possible use for wider and more different portfolios could have been examined.

**Clarity:**

Some part could have been more detailed. The overall presentation is clear.


**Overall Review:**

Pro :

The study's examination of the trade-off between the flexibility provided by a vast portfolio and the difficulty of selecting a subset of relevant algorithms is intriguing, and the proposed approach is relevant, even if not entirely novel.

The use of meta-representation of algorithm performance provides an innovative way of achieving AAS.

The experiments are thoroughly detailed, and the results are comprehensively analyzed, providing valuable insights for practitioners.

The graph representation utilized in this study provides interesting visualizations of the resulting portfolios, offering opportunities for a better understanding of the portfolio's building process.

The discovery that portfolios customized to individual preferences using Shapley meta-representations offer superior performance and greater flexibility is a noteworthy finding.


Cons :

The initial presentation of the problem could have been more comprehensive, providing more detailed background information and context.

While the state of the art is briefly discussed, a more thorough analysis could have been beneficial to contextualize the proposed approach further.

Despite being addressed, the computational cost induced by the proposed approach remains a significant limitation that requires careful consideration.

Experiments are limited to one kind of algorithm. The performance scope could be biased by this restriction.


**Potential Impact On The Field Of Automl:**

The paper's focus is primarily on BBO practitioners and may not be directly applicable to those working on ML tasks, even though AAS is a machine learning-based process. It is worth noting that the HPO task considered in this study involves selecting variants of CMA-ES, which forms the foundation of the portfolios. This task can be viewed as an AutoML problem, which is relevant to both the BBO and ML communities.

**Review Confidence:**

4: You are confident in your assessment, but not absolutely certain. It is unlikely, but not impossible, that you did not understand some parts of the submission or that you are unfamiliar with some pieces of related work.

**Review Rating:**

6: Borderline Leaning Accept: Technically sound paper where reasons to accept outweigh reasons to reject. Please use sparingly.

**Review Summary:**

The main contribution of this paper is the assessment of two meta-representation techniques, namely SHAP and performance2vec, to choose portfolios from CMA-ES variations for BBOB. Two types of portfolios have been considered: those related to the algorithm's overall behavior and those personalized to the algorithm's behavior for each problem. The experiments show that Performance2vec-based portfolios favor smaller sizes with minimal error, while SHAP-based portfolios are more adaptable but have lower performance. Personalized portfolios offer comparable or marginally better outcomes in nearly all situations compared to the classical greedy approach of selecting the best-performing algorithms based on reference tasks.

The paper focuses on BBO and specifically addresses the AAS problem for optimization, which is a well-known and extensively researched topic. The proposed methodology is highly pertinent, and the results obtained are intriguing for the BBO optimization community. Although some sections could have benefited from additional elaboration, the paper is well-written overall.


**Technical Quality And Correctness:**

Some technical details are missing. The experiments are clearly described. The experimental process is made available on the web for reproducibility purposes.

---

> ### Author Response · Authors · 2023-05-01
> **Response to Reviewer C1SQ**
>
> We thank you for your helpful and valuable comments. In the following, we address your individual comments. Please refer to the newly updated version of the paper for modifications.
>
> ### Limited impact ###
>
> Investigating the use of our approach on a wider range of inherently different algorithms is certainly one of the highest-priority TODOs in the future. However, the algorithm selection task itself is complex enough when we deal with a large family of similar algorithms, such as the modular CMA-ES framework. To this end, we believe that the essence of the observations made in this paper will propagate across different considered portfolios, meaning that the data-driven techniques for portfolio selection carry untapped potential for further performance gains.
>
> ### Initial problem presentation/SOTA ###
>
> We revised the introduction and added a paragraph on related work. Please note there is a surprising lack of works tackling this topic.
>
> ### Computational cost ###
>
> We do agree with the comment about the computational cost, which is one of the drawbacks listed in Section 5. We believe that, going forward, we can keep the computational cost low by reusing the explanations stemming from other studies on automated algorithm selection/configuration and making them more widely interpretable by including them in a semantic model such as the OPTION ontology.

---

### Official Review · Reviewer_i5Kx · 2023-04-13

**Potential Impact On The Field Of Automl Rating:** 3
**Technical Quality And Correctness Rating:** 3
**Clarity Rating:** 3

**Summary Of Contributions:**

The authors evaluate how the performance of automated algorithm selection (AAS) depends on the portfolio of algorithms to choose from. There exists a tradeoff in the size of the portfolio: a larger portfolio contains the best algorithm with higher probability, but selecting the best algorithm out of a large number of candidates is also harder. The authors propose constructing a similarity graph from a set of algorithms based on their meta-representations, computed using performance2vec or Shapley values (SHAP). Then, given a certain constraint on the portfolio size, graph algorithms like Maximal Independent Sets (MS) and Dominating Sets (DS) can be leveraged to select a portfolio that is representative and nonredundant.

The authors conclude that performane2vec selects a smaller portfolio but achieves better AAS performance relative to the true optimal algorithm, while SHAP selects a larger portfolio but achieves worse AAS performance. The authors further conclude that the personalized method, where portfolios are constructed for each problem, generally outperforms other baselines.

All of the proposed methods outperform the case of having a full portfolio.


**Actions Required To Increase Overall Recommendation:**

- clarity issues mentioned above
- adding a section on related work


**Clarity:**

The methodology in section 2 could be explained more clearly, either with more precise / consistent language or with visual / notational aid. I find myself confused and inferring a lot about what the authors meant when reading the paper.
The authors could consider having a figure illustrating the difference in performance2vec and SHAP, or having formal mathematical / algorithmic notation that distinguishes the different problem classes, problem instances, replications, and different problem features.
When describing performance2vec, the authors said "each vector element is the mean value of the quality of the solution of multiple problem instances on a particular problem", but also that "the quality of the solution of a given problem instances is computed as the median across multiple runs of the algorithm on that particular problem instance". If I understand correctly, there is a nested for loop here, where the inner loop is over running an algorithm multiple times on one problem instance and the outer loop is over different problem instances for a problem. If so, it could be nice to define those terms clearly.
Does "data instance" (line 97) indicate the algorithms in both the non-personalized and personalized case?
If I understand correctly, performance2vec produces algorithm representations that are n-dimensional (where n is the number of problem instances) and SHAP produces algorithm representations that are p-dimensional (where p is the number of landscape features). The authors did indicate that later in the experiment section, but it would be nice to state that clearly when presenting the methodology.

Section 3:
What does the "empirical cumulative distribution function" refer to?

Figure 4 is a very nice illustration of the tradeoff between the quality of the optimal algorithm in the portfolio and the flexibility of the portfolio.


**Overall Review:**

The methodology is valid and innovative. The experiments are well-designed and thoroughly conducted and the insights drawn from the experiments are explained well (e.g., Figure 4).

The methodology could be improved in terms of clarity of explanation. The paper may also benefit from adding a section on related work.


**Potential Impact On The Field Of Automl:**

The methodology is well-designed and appropriate for addressing the research questions. The results are interesting and provide new insights into how the portfolio affects AAS performance, which could have important implications for the AAS field. Others are likely to benefit from this paper's insights and cite this paper for its contributions.


**Review Confidence:**

3: You are fairly confident in your assessment. It is possible that you did not understand some parts of the submission or that you are unfamiliar with some pieces of related work.

**Review Rating:**

7: Weak Accept: Technically sound paper with moderate-to-high impact and strong evaluation, with perhaps some minor flaws.

**Review Summary:**

The methodology is valid and innovative, and the experimental results demonstrate better behavior than other baselines. The authors also provide insight into interpreting the experimental results.There are some issues with clarity mentioned above. Therefore I recommend weak accept.


**Technical Quality And Correctness:**

The proposed methodology is reasonable. The experiments conducted and conclusions drawn seem to be sound and of high quality.

One question in Section 4:
"For problems such as 7, 19, and 24, personalized portfolios exhibit the overall best performance" (line 261-262). Looking at figure 3, it seems to me that this is not the case. The green box does not lie significantly above the other ones for these problems. Would appreciate it if the authors could help clarify that.

---

> ### Author Response · Authors · 2023-05-01
> **Response to Reviewer i5Kx**
>
> We thank you for your helpful and valuable comments. In the following, we address your individual comments. Please refer to the newly updated version of the paper for modifications.
>
> ### Question on Section 4 ###
>
> We do agree there is no significant difference between the personalized portfolios boxplot and the greedy and full portfolio ones; we removed this claim to steer clear of further confusion.. However, on some problems we actually do observe a slight edge in favor of personalized portfolios across all considered dimensions and budgets (please refer to our Zenodo repo for all figures).
>
> ### Clarity issues and related work ###
>
> We revised the methodology section by explaining everything more clearly and by introducing mathematical notation to support the storyline. The “nested loop” of the p2v paragraph is now rewritten to increase accessibility. We added a reference for the empirical cumulative distribution function (ECDF); by definition, the value of ECDF at any specified value of the measured variable is the fraction of observations of the measured variable that are less than or equal to the specified value. In our experiments, area under ECDF curve is the performance of the algorithm up to a certain point. We also added a paragraph on related work in the Introduction section.

---

### Official Review · Reviewer_ZKBW · 2023-04-13

**Potential Impact On The Field Of Automl Rating:** 3
**Technical Quality And Correctness Rating:** 3
**Clarity Rating:** 4

**Summary Of Contributions:**

In this work, the authors investigate how the makeup of an algorithm portfolio affects automated algorithm selection (AAS) from that portfolio. In particular, they explore meta-representation (SHAP and performance2vec) driven selection techniques for building such a portfolio. They ultimately show discrepancies in portfolios selected based on each meta-representation approach, and qualitative differences in AAS performance.

**Actions Required To Increase Overall Recommendation:**

Ideally, develop a new meta-representation learning method that avoids the need to learn an instance-specific portfolio.

**Clarity:**

The writing is clear and easy to follow. Some specific comments:

- In Figure 2, it would be better to compare normalized/percentage difference in total loss
- The bottom row of Figure 1 could be reworked -- it is not very informative in its current state

**Overall Review:**

Strengths
- New and interesting perspective on data-driven portfolio selection
- Personalized version of their algorithm demonstrates better performance over existing greedy selection paradigm

Weaknesses
- No novel contribution on the meta-representation learning side
- P2V and SHAP serve as reasonable baselines, but it's been shown many times that SHAP is not a robust explainability method -- I'm confident that its feature attributions serve as a grounded source of truth here either
- Non-personalized versions of their method don't improve upon baseline

**Potential Impact On The Field Of Automl:**

The authors introduce an alternative perspective on the portfolio construction problem, but do not provide any novel algorithms of their own (in terms of meta-representation learning). However, their results do suggest that their method can indeed lead to somewhat better ASS performance.

**Review Confidence:**

3: You are fairly confident in your assessment. It is possible that you did not understand some parts of the submission or that you are unfamiliar with some pieces of related work.

**Review Rating:**

7: Weak Accept: Technically sound paper with moderate-to-high impact and strong evaluation, with perhaps some minor flaws.

**Review Summary:**

Overall, this work presents an interesting approach to portfolio selection. While some of the experimental results are not very convincing, and the choice of meta-representation learning methods is not perfect (and there is also no work done on novel meta-representation learning methods), the idea itself is worth introducing to the community.

**Technical Quality And Correctness:**

The approach is sound, albeit each component of their approach is not particularly novel.

---

> ### Author Response · Authors · 2023-05-01
> **Response to Reviewer ZKBW**
>
> We thank you for your helpful and valuable comments. In the following, we address your individual comments. Please refer to the newly updated version of the paper for modifications.
>
> ### No novelty ###
>
> The design of meta-representations of optimization algorithms is not within the scope of this paper. It is a complex task that merits its own research; we plan to focus substantial research efforts towards this goal. The designed meta-representations must be evaluated in detail and demonstrated to work well before being used in downstream applications such as ours. For this reason, we have chosen to rely on previously proposed meta-representations and demonstrate how they can be used to select complementary algorithms for an algorithm portfolio. However, the suggested approach is general and can be applied using any meta-representations developed in the future.
>
> ### Shapley values ###
>
> Shapley values quantify individual contribution of input features to the overall performance of the ML model. These contributions capture essential information about what can be thought of algorithm-instance interaction, both on a local and a global level. For these reasons, we believe that the SHAP method can be effectively used as a possible algorithm meta-representation, despite being instance-specific.
>
> ### Figures ###
>
> Bottom row of Figure 1 unfortunately depends on the visualization library we used (networkx, v2.5). We keep this in mind for the camera-ready version, until which we will have more time to explore different visualization options and choose one that works better. Thank you for pointing this out!